# Health-related quality of life in non-alcoholic fatty liver disease: A cross-cultural study between Spain and the United Kingdom

Jesús Funuyet-Salas[1]*, Agustín Martín-Rodríguez[2], María Ángeles Pérez-San-Gregorio[2], Luke Vale[3,4], Tomos Robinson[3], Quentin M. Anstee[5,6‡], Manuel Romero-Gómez[7‡]

1 Department of Psychology, Loyola University, Seville, Spain, 2 Faculty of Psychology, Department of Personality, Assessment, and Psychological Treatment, University of Seville, Seville, Spain, 3 Faculty of Medical Sciences, Population Health Sciences Institute, Health Economics Group, Newcastle University, Newcastle upon Tyne, United Kingdom, 4 National Institute for Health Research (NIHR) Newcastle In Vitro Diagnostics Co-Operative and NIHR Applied Research Collaboration North East and North Cumbria, Newcastle University, Newcastle upon Tyne, United Kingdom, 5 Faculty of Medical Sciences, Translational & Clinical Research Institute, Newcastle University, Newcastle upon Tyne, United Kingdom, 6 Newcastle NIHR Biomedical Research Centre, Newcastle upon Tyne Hospitals NHS Trust, Newcastle upon Tyne, United Kingdom, 7 Institute of Biomedicine of Seville, UCM Digestive Diseases and Ciberehd, Virgen del Rocío University Hospital, University of Seville, Seville, Spain

‡ QMA and MRG are Joint Senior Authors.
* jfunuyet1@us.es

**Data Availability Statement:** Data cannot be shared publicly because the Ethics Committee of

## Abstract

### Background

It is unclear what biopsychosocial factors influence the impact of NAFLD on health-related quality of life (HRQoL), and if these factors are equally important predictors between different nationalities.

### Methods

HRQoL (CLDQ) was measured in both Southern European (Spain, n = 513) and Northern European (United Kingdom -UK-, n = 224) cohorts of patients with NAFLD in this cross-sectional study. For each cohort, participant data were recorded on histological grade of steatohepatitis, stage of fibrosis and biopsychosocial variables. Regression analysis was used to explore which of these variables predicted HRQoL. Moderated mediation models were conducted using SPSS PROCESS v3.5 macro.

### Results

Participants with severe fibrosis reported more fatigue, systemic symptoms and worry, and lower HRQoL than those with none/mild fibrosis, regardless of place of origin. In addition, body mass index (BMI) and gender were found to be significant predictors of HRQoL in both Spanish and UK participants. Female gender was associated with worse emotional function, higher BMI and more fatigue, which predicted lower participants' HRQoL. UK participants showed more systemic symptoms and worry than Spanish participants, regardless of liver

the Virgen del Rocío University Hospital of Seville and the NHS HRA North East –Tyne & Wear South Research Ethics Committee have imposed restrictions on sharing data set for ethical reasons of privacy and confidentiality. Data are available from the LITMUS Study Cohort of the European NAFLD Registry (contact via all@litmus-project.es) for researchers who meet the criteria for access to confidential data. The data underlying the results presented in the study are available from all@litmus-project.es.

**Funding:** This study was funded by the Liver Investigation: Testing Marker Utility in Steatohepatitis (LITMUS) consortium which is funded by the Innovative Medicines Initiative (IMI2) Program of the European Union under Grant Agreement 777377, which receives funding from the EU Horizon 2020 programme and European Federation of Pharmaceutical Industries and Associations (EFPIA). The European NAFLD Registry and the Newcastle NIHR Biomedical Research Centre provided support so that this project could be carried out in Newcastle, UK. This study was also funded by the Fondo Europeo de Desarrollo Regional (FEDER)/Ministerio de Ciencia e Innovación—Agencia Estatal de Investigación in the form of a grant to JF-S, MAP-S-G, and AM-R [project PSI2017-83365-P], the Ministerio de Educación y Formación Profesional in the form of a grant to JF-S [project FPU16/03146], and the Gilead Sciences, Inc. in the form of an unrestricted grant to MR-G; this funding was provided so that this study could be carried out in Spain.

**Competing interests:** The authors have read the journal's policy and have the following potential competing interests: The Liver Investigation: Testing Marker Utility in Steatohepatitis (LITMUS) consortium is funded by the Innovative Medicines Initiative (IMI2) Program of the EU which receives funding from the European Federation of Pharmaceutical Industries and Associations (EFPIA). There are no patents, products in development or marketed products associated with this research to declare. This does not alter our adherence to PLOS ONE policies on sharing data and materials.

severity. The negative effects of gender on HRQoL through emotional function, BMI and fatigue were reported to a greater degree in UK than in Spanish participants.

## Conclusions

UK participants showed a greater impairment in HRQoL as compared to Spanish participants. Higher fibrosis stage predicted lower HRQoL, mainly in the Spanish cohort. Factors such as female gender or higher BMI contributed to the impact on HRQoL in both cohorts of patients and should be considered in future multinational intervention studies in NAFLD.

## Introduction

The number of people diagnosed with chronic non-communicable diseases around the world continues to rise [1]. Among these is non-alcoholic fatty liver disease (NAFLD), which in the 21st century has become one of the world's main causes of liver disease and liver transplant. NAFLD includes a spectrum of metabolic liver pathologies which go from simple hepatic steatosis to non-alcoholic steatohepatitis (NASH), accumulating fibrosis, cirrhosis and hepatocarcinoma. NAFLD is considered the liver manifestation of metabolic syndrome, with obesity identified as its main and most common risk factor. There is a close two-way relationship between the two pathologies [2].

With respect to the clinical impact of NAFLD, fibrosis has been established as an important predictor of patient mortality [3]. Predictive models for prognosis and survival, such as the MELD (Model for End-Stage Liver Disease) score, have been developed. This scale, based on International Normalized Ratio (INR) for prothrombin time and serum bilirubin and creatinine levels, is a reliable measure of mortality risk in patients with end-stage liver disease. Its use as a measure of liver function is generalisable to patient populations of diverse etiologies and wide ranges of severity [4].

However, until recently the impact of NAFLD from the patient's viewpoint had not been assessed. The increasing use of patient-reported outcomes (PROs) allows attention not just on the prevention and treatment of disease symptoms, but on the individual's physical, mental and social functioning and well-being—this is referred to as health-related quality of life (HRQoL) [5]. Several PRO measures have been used to assess the impact of NAFLD from a patient's point of view on their HRQoL and illness experience, most notably the Chronic Liver Disease Questionnaire (CLDQ). CLDQ is a liver disease-specific instrument which evaluates changes in physical and mental HRQoL due to liver disease. It addresses problems commonly reported by these patients such as fatigue or physical symptoms, as well as the mental or emotional impact of the disease. Higher scores indicate better HRQoL [6]. In fact, it has been shown that NAFLD impacts HRQOL mainly through physical health and activities of daily living [5,7]. Some factors contributing to reduced HRQoL are fatigue or lack of energy, daytime somnolence, abdominal pain or general pain [8]. NAFLD is also associated with significant mood disturbance, especially an increase in depression symptoms, which may also explain the impairment of the patient's well-being [9].

The evidence to date on the effect of NASH and fibrosis on the HRQoL of NAFLD patients is inconsistent [3,10–13]. NASH has been associated with worse HRQoL, primarily in physical aspects of patients' well-being [14]. NASH has even been linked to an overall impairment in HRQoL in a recent study using symptom elicitation and cognitive debriefing interviews [15]. Although when controlling for other factors, it has been shown that there is no evidence of an association [10,11]. The evidence for fibrosis being a predictor of HRQoL in NAFLD is mixed.

Some researchers have reported evidence of an inverse relationship between the severity of fibrosis and HRQoL [10,11] whilst others have found no evidence of an association [3,12]. Obesity has likewise been reported as reducing HRQoL [12,16–18], however other studies have not provided any evidence of such a relationship [13,19]. There is more consistency with respect to impact of gender on HRQoL for those with NAFLD, with females with NAFLD reporting a greater decrement on physical and mental functioning compared with males [5,10–12]. Lastly, the influence on HRQoL of other sociodemographic factors such as age [5,10,19], education [10,11,17] or employment status [10,11,20] have also been investigated, but there is no conclusive evidence of an impact to date.

Cross-cultural research has been widely recommended in the field of health care, since the illness experience may vary according to the socio-cultural context in which the person has developed [21]. It would be important to understand how the impact of NAFLD on patients' HRQoL varies according to their place of origin, especially in order to consider these differences in future multinational intervention and treatment-effectiveness studies in NAFLD. Only one study has compared the HRQoL of NAFLD patients in different European countries [12]. This study compared the United Kingdom (UK) and Germany and found a substantial burden of symptoms in patients, especially in UK, with variables such as age, sex or lobular inflammation correlating with lower HRQoL. Given the limited data currently available, and that biopsychosocial factors influencing and predicting HRQoL in NAFLD patients remain unclear, the current study seeks to further explore whether there are geographic variations in how NAFLD affects HRQoL. This paper therefore compares two patient cohorts: one from Spain and one from the UK. Specifically, we addressed three primary objectives: 1) to compare HRQoL of NAFLD patients based on place of origin (Spain or UK) and severity of liver disease (absence or presence of NASH, and fibrosis stage); 2) to identify what histological and biopsychosocial variables predict HRQoL in Spanish and UK patient cohorts; and 3) to analyse what biopsychosocial variables mediated or moderated in HRQoL predictive models.

## Material and methods

### Participants and study sample

The sample comprised 737 biopsy-proven NAFLD patients. 513 participants were from Spain (HEPAmet Registry) and 224 from UK (European NAFLD Registry) [22]. Full details of participant sociodemographic characteristics may be seen in Tables 1 and 2.

All participants gave written informed consent for participation in the study, which was approved by the Ethics Committee of the Virgen del Rocío University Hospital of Seville (19/02/2017/EHGNA) for the Spanish cohort and NHS HRA North East–Tyne & Wear South Research Ethics Committee for the UK cohort (NCT04442334) [22]. The study was carried out in compliance with the Helsinki Declaration of 1975.

The 737 participants were consecutive prospectively recruited from 12 Spanish hospitals and 11 UK hospitals. All participants spoke the local language (Spanish or English) as their native tongue and were evaluated with a psychosocial interview and the CLDQ. To be included in the study, the participants had to be 18 years of age, give their informed consent for participating, have been diagnosed by liver biopsy as having NAFLD, show adequate understanding of the study evaluation instrument and not have a severe or disabling psychopathological condition.

The participants were classified by place of origin ($G_1$ = Spain, $G_2$ = UK), and by descriptors of severity of disease: NASH ($G_3$ = absence, $G_4$ = presence) and fibrosis ($G_5$ = none/mild, $G_6$ = moderate, $G_7$ = severe) (Fig 1). NASH was determined by a value of activity greater than or equal to 2 as the SAF (Steatosis, Activity and Fibrosis) score [23]. Fibrosis was categorized as none/mild (stages F0 and F1), moderate (F2 and F3) or severe (F4, cirrhosis). The MELD score for each

**Table 1. Comparison of sociodemogaphic and clinic variables by place of origin (Spain and UK).**

| | Place of origin | | Intergroup comparisons | Effect sizes |
|---|---|---|---|---|
| | Spain (G$_1$) $n = 513$ | UK (G$_2$) $n = 224$ | | |
| | M (SD) | M (SD) | t (p) | Cohen's d |
| Age | 55.04 (11.83) | 55.31 (12.34) | $t_{(1,735)} = -0.281$ (0.779) | -0.022 N |
| BMI | 30.62 (5.12) | 34.85 (5.54) | $t_{(1,405.762)} = -9.681$ (<0.001) | -0.793 M |
| MELD score | 7.11 (1.81) | 6.93 (1.45) | $t_{(1,563)} = 1.239$ (0.216) | 0.110 N |
| | % | % | $\chi^2$ (p) | Cohen's w |
| Gender | | | $\chi^2_{(1)} = 2.246$ (0.134) | -0.055 N |
| 1. Male | 58.9 | 64.7 | | |
| 2. Female | 41.1 | 35.3 | | |
| Education | | | $\chi^2_{(1)} = 26.876$ (<0.001) | 0.194 S |
| i. Primary/Secondary | 73.5 | 53.3 | | |
| ii. Higher | 26.5 | 46.7 | | |
| Employment | | | $\chi^2_{(1)} = 7.510$ (0.006) | -0.102 S |
| iii. Actively employed | 47.6 | 58.8 | | |
| iv. Not actively employed | 52.4 | 41.2 | | |
| Liver fibrosis | | | $\chi^2_{(1)} = 96.894$ (<0.001) | 0.363 M |
| v. None/mild | 62.2 | 22.8 | | |
| vi. Moderate or severe | 37.8 | 77.2 | | |

Effect sizes: N, null; S, small; M, medium. The t-test for independent samples was applied for continuous variables. Pearson's Chi-square was applied for categorical variables.

participant was calculated as a marker of hepatic function. This score was calculated as a measure of severity of liver impairment based on three laboratory parameters: INR for prothrombin time and serum bilirubin and creatinine [4]. Other factors used to describe the participants were age, body mass index (BMI), gender (male or female), education (primary, secondary or higher education) and employment status (actively employed or not actively employed).

## Health-related quality of life assessment

HRQoL was measured using the CLDQ [6]. This instrument includes 29 items with seven-point Likert-type scales on the following HRQoL dimensions: abdominal symptoms, activity, emotional function, fatigue, systemic symptoms, and worry. It also provides a total score corresponding to the mean of the scores on each of the dimensions. All scores range from 0 (worst HRQoL) to 7 (best HRQoL). In terms of internal consistency, in the total sample the Cronbach's alpha [24] was 0.95 for the total score and ranged from 0.65 to 0.89 for the different dimensions. For the Spanish cohort the alpha was 0.92 for the total score and ranged from 0.78 to 0.93 for the different dimensions. For the UK cohort it was 0.96 for the total score and ranged from 0.78 and 0.93 for the different dimensions.

## Statistical analysis

The following were used for between-group comparisons of the sociodemographic and clinical variables: an independent samples t-test or one-way ANOVA (Welch´s U or Snedecor's F) with Games-Howell or Tukey HSD post hoc pairwise analysis for continuous variables (age, body mass index and MELD score), and Pearson's chi-square test for categorical variables

**Table 2. Comparison of sociodemogaphic and clinic variables by NASH (absence and presence) and fibrosis (none/mild, moderate and severe).**

| | NASH | | | Intergroup comparisons | Effect sizes |
|---|---|---|---|---|---|
| | **Absence (G$_3$)** $n = 331$ | **Presence (G$_4$)** $n = 406$ | | | |
| | *M (SD)* | *M (SD)* | | *t (p)* | **Cohen's d** |
| Age | 54.30 (12.38) | 55.80 (11.61) | | $t_{(1,375)}$ = -1.692 (0.091) | -0.125 N |
| BMI | 30.69 (5.50) | 32.98 (5.49) | | $t_{(1,704)}$ = -5.515 (<0.001) | -0.417 S |
| MELD score | 7.06 (1.79) | 7.02 (1.60) | | $t_{(1,563)}$ = 0.288 (0.773) | 0.023 N |
| | % | % | | $\chi^2 (p)$ | **Cohen's w** |
| Gender | | | | $\chi^2_{(1)}$ = 0.116 (0.734) | 0.013 N |
| • 1. Male | 61.3 | 60.1 | | | |
| • 2. Female | 38.7 | 39.9 | | | |
| Education | | | | $\chi^2_{(2)}$ = 15.399 (<0.001) | 0.147 S |
| • Primary/Secondary | 71.6 | 64.7 | | | |
| • Higher | 28.4 | 35.3 | | | |
| Employment | | | | $\chi^2_{(1)}$ = 1.530 (0.216) | 0.046 N |
| • Actively employed | 53.4 | 48.7 | | | |
| • Not actively employed | 46.6 | 51.3 | | | |
| | Fibrosis | | | Intergroup comparisons | Effect sizes |
| | **None/mild (G$_5$)** $n = 370$ | **Moderate (G$_6$)** $n = 286$ | **Severe (G$_7$)** $n = 81$ | | |
| | *M (SD)* | *M (SD)* | *M (SD)* | *U/F (p)* | **Cohen's d** |
| Age | 52.60 (12.48) | 56.63 (11.08) | 61.35 (9.39) | $U_{(2,245.602)}$ = 26.975 (<0.001) | |
| | | | | G$_5$-Gb$_6$ (<0.001) | -0.341 S |
| | | | | G$_5$-Gb$_7$ (<0.001) | -0.792 M |
| | | | | G$_6$-Gb$_7$ (0.001) | -0.459 S |
| BMI | 30.72 (5.27) | 33.33 (5.64) | 32.80 (5.83) | $F_{(2,703)}$ = 18.622 (<0.001) | |
| | | | | G$_5$-Gb$_6$ (<0.001) | -0.478 S |
| | | | | G$_5$-Gb$_7$ (0.007) | -0.374 S |
| | | | | G$_6$-Gb$_7$ (0.729) | 0.092 N |
| MELD score | 6.89 (1.72) | 6.96 (1.37) | 7.78 (2.23) | $U_{(2,182.761)}$ = 5.011 (0.008) | |
| | | | | G$_5$-Gb$_6$ (0.883) | -0.045 N |
| | | | | G$_5$-Gb$_7$ (0.006) | -0.447 S |
| | | | | G$_6$-Gb$_7$ (0.010) | -0.443 S |
| | % | % | % | $\chi^2 (p)$ | **Cohen's w** |
| Gender | | | | $\chi^2_{(2)}$ = 2.437 (0.296) | 0.058 N |
| • Male | 62.4 | 60.5 | 53.1 | | |
| • Female | 37.6 | 39.5 | 46.9 | | |
| Education | | | | $\chi^2_{(4)}$ = 5.063 (0.281) | 0.084 N |
| • Primary/Secondary | 67.9 | 66.5 | 72.2 | | |
| • Higher | 32.1 | 33.5 | 27.8 | | |
| Employment | | | | $\chi^2_{(2)}$ = 21.036 (<0.001) | 0.170 S |
| • Actively employed | 59.0 | 43.9 | 36.8 | | |
| • Not actively employed | 41.0 | 56.1 | 63.2 | | |

Effect sizes: N, null; S, small; M, medium. The t-test for independent samples or one-way ANOVA (Welch´s *U* / Snedecor's *F*) with Games-Howell / Tukey *HSD* post-hoc pairwise analysis were applied for continuous variables. Pearson's Chi-square was applied for categorical variables.

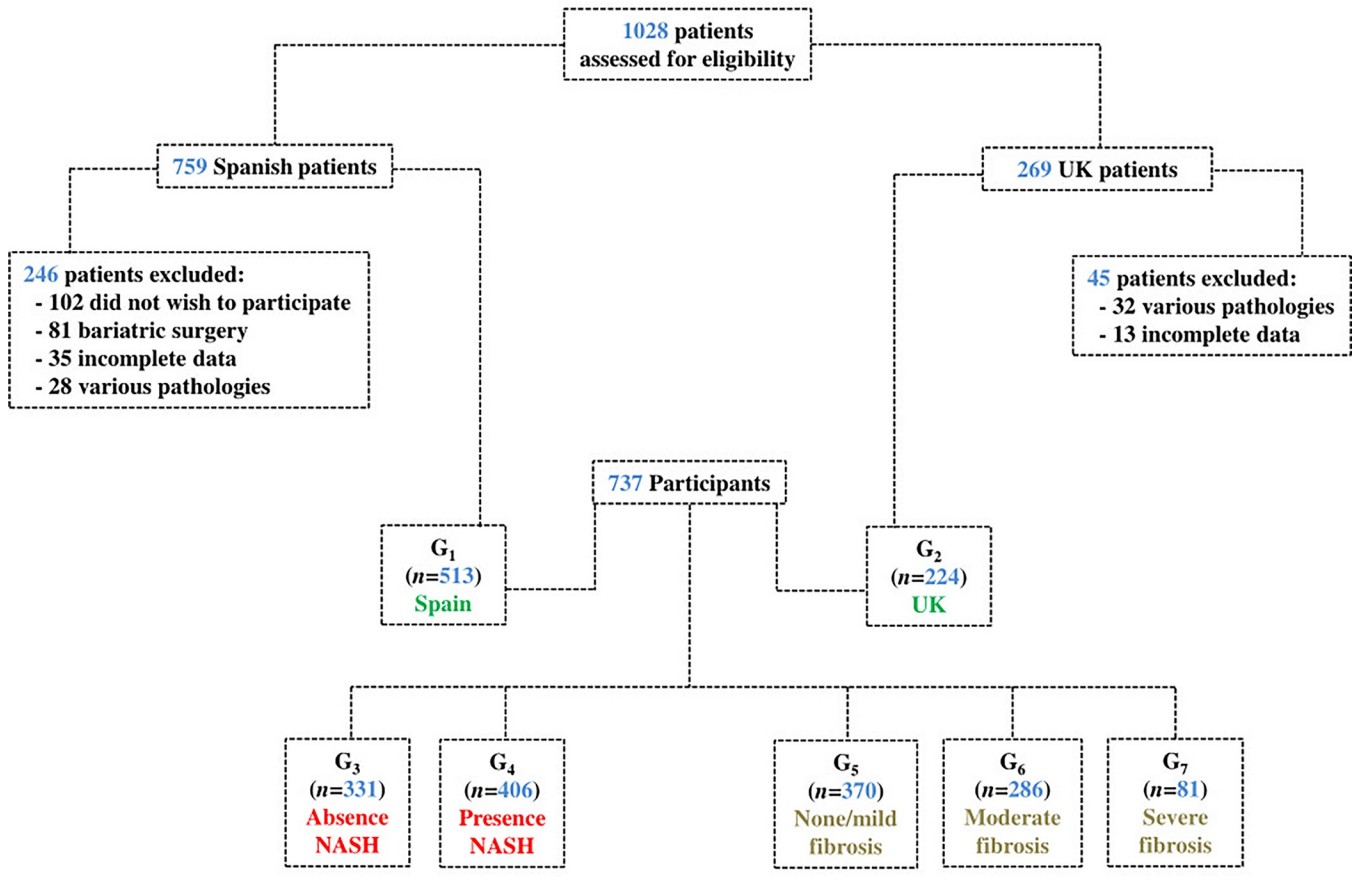

**Fig 1. Participant selection for the study.**

(gender, education, employment status and liver fibrosis). Categorical variables were dichotomised into: male or female gender, primary/secondary or higher education, active or non-active employment status, and none/mild or moderate or severe fibrosis. Cohen's *d* (for continuous variables) and *w* (for categorical variables) were computed as effect size indexes. Effect sizes are defined as: null ($d < 0.2$; $w < 0.1$), small ($d \geq 0.2$; $w \geq 0.1$), medium ($d \geq 0.5$; $w \geq 0.3$) or large ($d \geq 0.8$; $w \geq 0.5$) [25]. Only statistically significant differences with medium or large effect sizes were considered important in this manuscript.

Missing values were imputed with SPSS Statistics v.25. Missing values were found for MELD score, education and employment status, but were less than 5% of the total data (1.1, 3.4 and 1.8%, respectively). Therefore, these values were assumed to be missing at random.

A 2x2 factorial ANOVA (Snedecor's *F*) was used to analyse the influence of place of origin (Spain or UK) and NASH (absence or presence) on HRQoL. To explore the influence of place of origin (Spain or UK) and fibrosis (none/mild, moderate or severe), a 2x3 factorial ANOVA (Snedecor's *F*) was applied.

A binary logistic regression analysis was used to determine the contribution of histological and biopsychosocial factors to HRQoL in both Spanish and UK participants separately. Nagelkerke's $R^2$/AIC/BIC was calculated as a goodness-of-fit measure. The accuracy index was calculated to check the percentage of cases correctly classified by the model. The independent variables in both regression models were NASH (absence or presence, which implied an activity score higher than or equal to 2 on the SAF score), fibrosis (none/mild fibrosis vs. moderate

or severe fibrosis), MELD score, BMI, gender (male or female), age, education (primary/secondary education only vs. higher education), and employment status (actively employed vs not actively employed). The reference categories for each variable were NASH, moderate or severe fibrosis, females, primary/secondary education, and not actively employed.

The dependent variable in both models was the total score on the CLDQ questionnaire (HRQoL). This score was arranged in ascending order and the cumulative percentages were used to divide both samples at the 50th percentile, forming two groups, one with better and the other with a worse HRQoL. The results of the binary logistic regression were presented as odds ratios (OR) with 95% confidence intervals. Those with a $p$-value below 0.05 were considered statistically significant. All data were analysed with SPSS Statistics v.25.

In order to identify what biopsychosocial variables mediated or moderated HRQoL in both patient cohorts, mediation and moderated mediation models were applied using the SPSS PROCESS macro v3.5 [26]. The CLDQ emotional function dimension was analysed to determine the role of mood in participants' perceived HRQoL. Fatigue was included as it is the main symptom associated with NAFLD [27] and because of its determinant role in our study, as demonstrated by the interactive effects found in the first objective. BMI and gender were also included as predictors of HRQoL, according to the results of our second objective. Thus, emotional function, BMI and fatigue were used as the mediators in the relationship between gender and HRQoL, applying Model 6. This is a mediation model in which the mediation effect of three variables on the relationship between the independent variable and the dependent one can be analysed [28]. Bootstrapping with 5000 resamples was used to test the estimated indirect effects. Mediation was considered significant if the 95% confidence interval (CI) of the indirect effects did not include 0. In continuation, Model 87 was applied. This is a moderated mediation model in which the moderating effect of one variable on a model with three mediating variables can be analysed [28]. 5000 bootstrap resamples were used to analyse the effect of moderated mediation, that is, whether the place of origin moderated the indirect effects of gender on the HRQoL through emotional function, BMI and fatigue. Moderation significance was tested and the conditional effect of the predictor on the criterion variable was calculated for each value of the moderator by generating its confidence interval [29]. Those with a $p$-value below 0.05 were considered statistically significant.

## Results

### Sociodemographic and clinical variables

The only important between-group differences (medium or large effect sizes) in sociodemographic and clinical variables (age, gender, education, employment status, BMI, liver fibrosis and MELD score) were that UK participants ($G_2$, $M = 34.85$, $SD = 5.54$) had a higher BMI than Spanish participants ($G_1$, $M = 30.62$, $SD = 5.12$) ($p < 0.001$, $d = -0.793$) (Table 1). UK participants also had a higher fibrosis stage ($G_2$, 77.2% had moderate or severe fibrosis) than Spanish participants ($G_1$, 37.8% had moderate or severe fibrosis) ($p < 0.001$, $d = 0.363$) (Table 1). Finally, participants with severe fibrosis ($G_7$, $M = 61.35$, $SD = 9.39$) were older than those with none/mild fibrosis ($G_5$, $M = 52.60$, $SD = 12.48$) ($p < 0.001$, $d = -0.792$) (Table 2).

### Objective 1. Influence of place of origin, NASH and fibrosis on health-related quality of life

**Interactive effects.** Table 3 shows HRQoL results by place of origin and NASH, while Table 4 shows HRQoL results by place of origin and fibrosis. The analyses provided evidence for two interactive effects: fatigue ($p = 0.003$, Table 4) and HRQoL ($p = 0.039$, Table 4). Simple

**Table 3. Health-related quality of life of NAFLD patients by place of origin (Spain and UK) and NASH (absence and presence).**

| CLDQ | Place of origin $M^a$ (SD) | | NASH $M^a$ (SD) | | Main effects | | Interaction effects |
|---|---|---|---|---|---|---|---|
| | Spain (G₁) $n = 513$ | UK (G₂) $n = 224$ | Absence (G₃) $n = 331$ | Presence (G₄) $n = 406$ | Place of origin $F_{(1,733)}$ $p$ (d) | NASH $F_{(1,733)}$ $p$ (d) | $F_{(1,733)}$ $p$ |
| Abdominal symptoms | 5.58 (1.58) | 5.42 (1.80) | 5.59 (2.18) | 5.41 (1.61) | 1.36 0.243 (0.094 N) | 1.50 0.221 (0.094 N) | 0.00 (0.967) |
| Activity | 5.69 (1.36) | 5.60 (1.65) | 5.78 (1.82) | 5.52 (1.41) | 0.55 0.460 (-0.007 N) | 4.27 0.039 (0.160 N) | 0.18 (0.675) |
| Emotional function | 5.71 (1.13) | 5.12 (1.50) | 5.50 (1.64) | 5.33 (1.21) | 29.74 <0.001 (0.444 S) | 2.46 0.117 (0.118 N) | 0.01 (0.907) |
| Fatigue | 5.31 (1.36) | 4.87 (1.80) | 5.27 (2.00) | 4.92 (1.41) | 10.75 0.001 (0.276 S) | 7.05 0.008 (0.202 S) | 0.00 (0.964) |
| Systemic symptoms | 5.88 (1.13) | 5.28 (1.20) | 5.71 (1.45) | 5.45 (1.01) | 37.85 <0.001 (0.515 M) | 7.43 0.007 (0.208 S) | 0.10 (0.753) |
| Worry | 6.11 (1.13) | 5.07 (1.35) | 5.64 (1.64) | 5.54 (1.21) | 91.54 <0.001 (0.835 L) | 0.72 0.397 (0.069 N) | 1.66 (0.198) |
| HRQoL | 5.71 (1.13) | 5.23 (1.20) | 5.58 (1.45) | 5.36 (1.01) | 26.76 <0.001 (0.412 S) | 5.36 0.021 (0.176 N) | 0.00 (0.958) |

[a] Higher scores show more health-related quality of life.

Effect sizes: N, null; S, small; M, medium; L, large. A 2×2 factorial ANOVA (Snedecor's $F$) was applied.

effects showed important effect sizes (medium or large) in Spanish participants (G₁) (Table 5 and Fig 2). In this respect, Spanish participants had more fatigue and lower HRQoL when they had severe fibrosis compared to those with moderate fibrosis (fatigue, $p = 0.001$, $d = 0.568$; HRQoL, $p = 0.001$, $d = 0.612$) or none/mild fibrosis (fatigue, $p < 0.001$, $d = 1.095$; HRQoL, $p < 0.001$, $d = 1.077$). Spanish participants with moderate fibrosis also suffered more fatigue than those with none/mild fibrosis ($p < 0.001$, $d = 0.552$).

Simple effects also showed important effect sizes (medium or large) in participants with none/mild fibrosis (G₅) (Table 5 and Fig 2). In this sense, participants with none/mild fibrosis suffered more fatigue ($p < 0.001$, $d = 0.566$) and lower HRQoL ($p < 0.001$, $d = 0.550$) if they were from the UK compared to Spanish participants.

**Health-related quality of life by place of origin.** In terms of the main effects, considering those with important effect sizes (medium or large), UK participants (G₂) had more systemic symptoms ($p < 0.001$, $d = 0.515$) and more worried ($p < 0.001$, $d = 0.835$) than Spanish participants (G₁), regardless of absence or presence of NASH (Table 3). UK participants (G₂) were more worried ($p < 0.001$, $d = 0.531$) than Spanish participants (G₁), no matter what the level of fibrosis was.

**Health-related quality of life by liver severity.** In terms of the main effects, considering those with important effect sizes (medium or large), participants with severe fibrosis (G₇) were more fatigued ($p < 0.001$, $d = 0.537$), had more systemic symptoms ($p < 0.001$, $d = 0.496$), more worried ($p < 0.001$, $d = 0.515$), and had a lower HRQoL ($p < 0.001$, $d = 0.642$) than those with none/mild fibrosis (G₅), regardless of place of origin (Table 4).

**Table 4. Health-related quality of life of NAFLD patients by place of origin (Spain and UK) and fibrosis (none/mild, moderate and severe).**

| CLDQ | Place of origin $M^a$ (SD) | | Fibrosis $M^a$ (SD) | | | Main effects | Interaction effects | |
|---|---|---|---|---|---|---|---|---|
| | Spain (G1) n = 513 | UK (G2) n = 224 | None/mild (G5) n = 370 | Moderate (G6) n = 286 | Severe (G7) n = 81 | Place of origin $F_{(1,731)}$ p (d) | Fibrosis $F_{(2,731)}$ p (d) | $F_{(1,733)}$ p |
| Abdominal symptoms | 5.26 (2.04) | 5.36 (1.80) | 5.68 (2.11) | 5.37 (1.52) | 4.88 (1.53) | 0.39 0.531 (-0.052 N) | 7.66 0.001 G5-Gb6 0.002 (0.168 N) G5-Gb7 <0.001 (0.434 S) G6-Gb7 0.028 (0.321 S) | 2.16 (0.116) |
| Activity | 5.42 (1.81) | 5.55 (1.50) | 5.89 (1.92) | 5.45 (1.35) | 5.12 (1.35) | 1.01 0.315 (-0.078 N) | 10.93 <0.001 G5-Gb6 <0.001 (0.265 S) G5-Gb7 <0.001 (0.464 S) G6-Gb7 0.112 (0.244 S) | 1.35 (0.260) |
| Emotional function | 5.49 (1.58) | 5.12 (1.35) | 5.64 (1.73) | 5.26 (1.18) | 5.01 (1.17) | 10.03 0.002 (0.252 S) | 9.33 <0.001 G5-Gb6 <0.001 (0.257 S) G5-Gb7 <0.001 (0.427 S) G6-Gb7 0.186 (0.213 S) | 1.89 (0.152) |
| Fatigue | 4.90 (2.04) | 4.73 (1.65) | 5.28 (2.11) | 4.87 (1.35) | 4.31 (1.44) | 1.58 0.209 (0.092 N) | 13.43 <0.001 G5-Gb6 <0.001 (0.231 S) G5-Gb7 <0.001 (0.537 M) G6-Gb7 0.004 (0.401 S) | 5.84 (0.003) |
| Systemic symptoms | 5.65 (1.36) | 5.18 (1.20) | 5.74 (1.54) | 5.42 (1.01) | 5.08 (1.08) | 20.72 <0.001 (0.366 S) | 12.05 <0.001 G5-Gb6 <0.001 (0.246 S) G5-Gb7 <0.001 (0.496 S) G6-Gb7 0.018 (0.325 S) | 1.61 (0.201) |
| Worry | 5.84 (1.58) | 5.06 (1.35) | 5.82 (1.73) | 5.48 (1.18) | 5.06 (1.17) | 46.85 <0.001 (0.531 M) | 12.27 <0.001 G5-Gb6 <0.001 (0.230 S) G5-Gb7 <0.001 (0.515 M) G6-Gb7 0.010 (0.357 S) | 1.51 (0.221) |
| HRQoL | 5.43 (1.36) | 5.17 (1.20) | 5.67 (1.35) | 5.31 (1.01) | 4.91 (0.99) | 7.09 0.008 (0.203 S) | 17.32 <0.001 G5-Gb6 <0.001 (0.302 S) G5-Gb7 <0.001 (0.642 M) G6-Gb7 0.004 (0.400 S) | 3.25 (0.039) |

a Higher scores show more health-related quality of life.

Effect sizes: N, null; S, small; M, medium. A 2×3 factorial ANOVA (Snedecor's $F$) was applied.

**Table 5. Simple effects in fatigue and total health-related quality of life.**

| Fibrosis | Spain | | UK | |
|---|---|---|---|---|
| | (G₁) | | (G₂) | |
| | n = 513 | | n = 224 | |
| | p | Cohen's d | p | Cohen's d |
| | Fatigue | | | |
| None/mild–Moderate | <0.001 | 0.552 M | 0.833 | 0.035 N |
| None/mild–Severe | <0.001 | 1.095 L | 0.204 | 0.265 S |
| Moderate–Severe | 0.001 | 0.568 M | 0.196 | 0.233 S |
| | HRQoL | | | |
| None/mild–Moderate | <0.001 | 0.485 S | 0.146 | 0.235 S |
| None/mild–Severe | <0.001 | 1.077 L | 0.046 | 0.430 S |
| Moderate–Severe | 0.001 | 0.612 M | 0.309 | 0.186 N |

| Place of origin | None/mild | | Moderate | | Severe | |
|---|---|---|---|---|---|---|
| | (G₅) | | (G₆) | | (G₇) | |
| | n = 370 | | n = 286 | | n = 81 | |
| | p | Cohen's d | p | Cohen's d | p | Cohen's d |
| | Fatigue | | | | | |
| Spain—UK | <0.001 | 0.566 M | 0.617 | 0.065 N | 0.241 | -0.259 S |
| | HRQoL | | | | | |
| Spain—UK | <0.001 | 0.550 M | 0.008 | 0.305 S | 0.631 | -0.101 N |

Effect sizes: N, null; S, small; M, medium; L, large.

**Objective 2. Histological and biopsychosocial predictors of health-related quality of life.** A binary logistic regression was used to evaluate the effect of the histological (NASH, fibrosis and MELD score) and biopsychosocial (BMI, gender, age, education and employment status) variables on HRQoL, both in Spanish (G₁) and UK (G₂) participants separately.

**1. Spanish cohort.** For Spanish participants, HRQoL reduced as fibrosis (OR = 0.290, 95% CI = 0.165–0.507, $p < 0.001$), MELD score (OR = 0.855, 95% CI = 0.744–0.982, $p = 0.027$) and BMI (OR = 0.921, 95% CI = 0.875–0.970, $p = 0.002$) increased. Lower HRQoL was also independently associated with female gender (OR = 0.297, 95% CI = 0.176–0.501, $p < 0.001$) (Table 6).

Nagelkerke's $R^2$ was calculated as a goodness-of-fit measure. The model explained 23.6% of the variance in QoL for the Spanish cohort, and 21.2% for the UK cohort.

For Spanish participants, the accuracy index was 0.702., therefore the model correctly classifies 70.2% of cases overall. Sensitivity was 75.3% and specificity 64.7%, while positive and negative predictive values were 0.699 and 0.706, respectively. For UK participants, the accuracy index was 0.681, therefore the model correctly classifies 68.1% of cases overall. Sensitivity was 69.5% and specificity 66.7%, while positive and negative predictive values were 0.680 and 0.681, respectively.

**2. UK cohort.** For UK participants, HRQoL reduced as BMI (OR = 0.942, 95% CI = 0.889–0.999, $p = 0.047$) increased. Lower HRQoL was also independently associated with female gender (OR = 0.448, 95% CI = 0.219–0.915, $p = 0.028$), non-active employment status (OR = 0.336, 95% CI = 0.152–0.745, $p = 0.007$) and younger age (OR = 1.065, 95% CI = 1.029–1.102, $p < 0.001$) (Table 6).

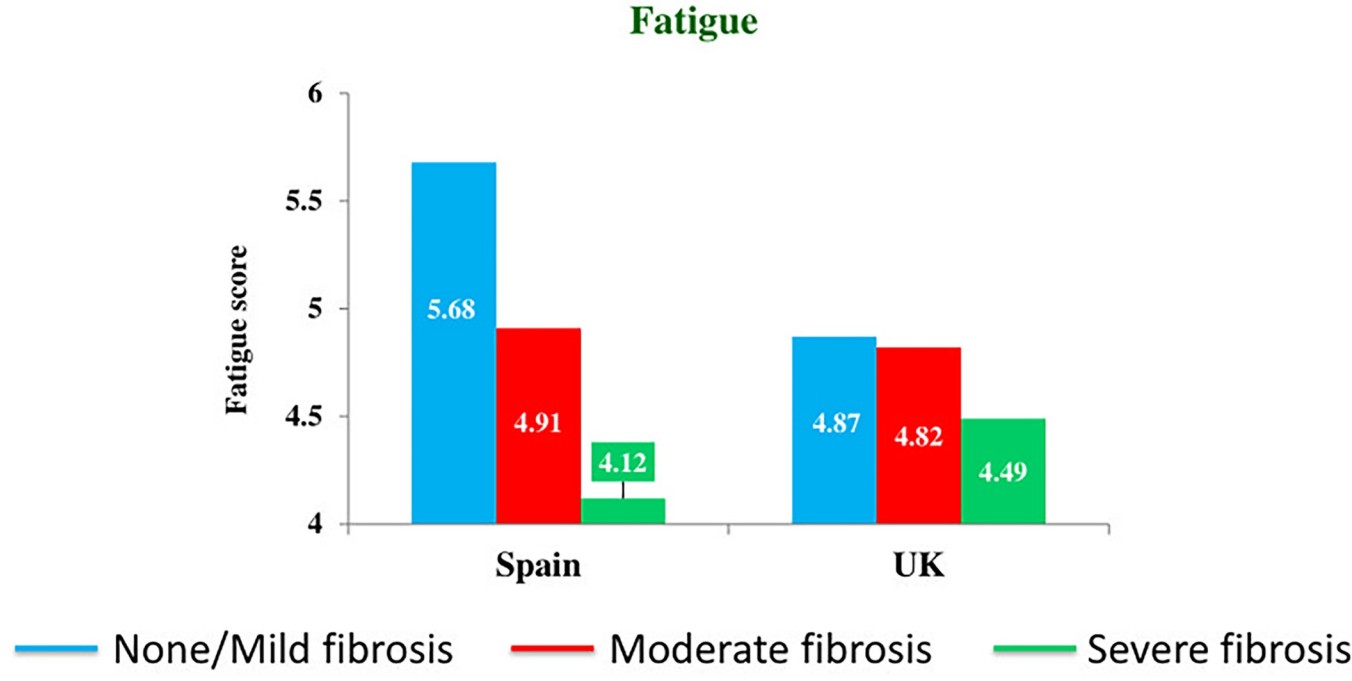

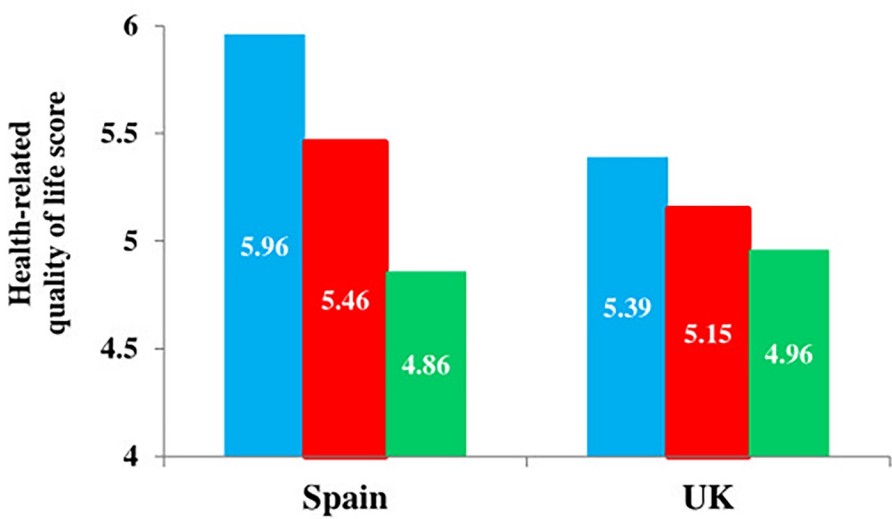

**Fig 2. Interactive effects of place of origin (Spain or UK) and fibrosis (none/mild, moderate or severe) factors.** Analysis of the influence of place of origin and fibrosis on the health-related quality of life of NAFLD patients showing interactive effects in fatigue ($p = 0.003$) and HRQoL ($p = 0.039$) (2x3 factorial ANOVA -Snedecor's $F$-). Scores vary from 1 to 7, higher scores showing better health-related quality of life.

### Objective 3. Mediation and moderated mediation analysis

**1. Mediation model.** Fig 3 and S1 Table show the relationships between the independent variable, the mediating variables and the dependent variable in the mediation model. In this model, the indirect effects of the emotional function, BMI and fatigue when mediating the relationship between gender and HRQoL can be tested. There was evidence for the following relationships: emotional function (effect = -0.200, $p < 0.001$); emotional function–BMI (effect

**Table 6. Binary logistic regression analysis with health-related quality of life as the dependent variable.**

| Spain | Coefficient | SE | AUC (CI) | p | OR | 95% CI | |
|---|---|---|---|---|---|---|---|
| | | | | | | Lower | Upper |
| NASH | 0.342 | 0.268 | 0.464 (0.414–0.514) | 0.202 | 1.408 | 0.833 | 2.381 |
| Fibrosis | -1.239 | 0.286 | 0.639 (0.578–0.699) | <0.001 | 0.290 | 0.165 | 0.507 |
| MELD score | -0.157 | 0.071 | 0.566 (0.504–0.628) | 0.027 | 0.855 | 0.744 | 0.982 |
| BMI | -0.082 | 0.026 | 0.601 (0.540–0.663) | 0.002 | 0.921 | 0.875 | 0.970 |
| Gender | -1.215 | 0.268 | 0.620 (0.559–0.681) | <0.001 | 0.297 | 0.176 | 0.501 |
| Age | 0.014 | 0.013 | 0.450 (0.400–0.500) | 0.251 | 1.015 | 0.990 | 1.040 |
| Education | 0.104 | 0.295 | 0.485 (0.435–0.535) | 0.725 | 1.109 | 0.622 | 1.979 |
| Employment | -0.224 | 0.287 | 0.573 (0.511–0.635) | 0.435 | 0.799 | 0.455 | 1.403 |
| **UK** | **Coefficient** | **SE** | **AUC (CI)** | **p** | **OR** | **95% CI** | |
| | | | | | | Lower | Upper |
| NASH | -0.045 | 0.415 | 0.519 (0.436–0.601) | 0.914 | 0.956 | 0.424 | 2.155 |
| Fibrosis | -0.403 | 0.426 | 0.524 (0.442–0.607) | 0.344 | 0.668 | 0.290 | 1.541 |
| MELD score | -0.154 | 0.130 | 0.482 (0.399–0.564) | 0.235 | 0.857 | 0.665 | 1.006 |
| BMI | -0.059 | 0.030 | 0.621 (0.541–0.701) | 0.047 | 0.942 | 0.889 | 0.999 |
| Gender | -0.803 | 0.364 | 0.583 (0.501–0.665) | 0.028 | 0.448 | 0.219 | 0.915 |
| Age | 0.063 | 0.017 | 0.614 (0.536–0.693) | <0.001 | 1.065 | 1.029 | 1.102 |
| Education | 0.267 | 1.229 | 0.510 (0.430–0.591) | 0.828 | 1.307 | 0.117 | 1.537 |
| Employment | -1.089 | 0.405 | 0.563 (0.481–0.645) | 0.007 | 0.336 | 0.152 | 0.745 |

SE, standard error; AUC, area under the ROC curve; OR, odds ratio; CI, confidence interval.

The logistic regression model was statistically significant for both Spanish ($\chi^2$ = 63.453, $p$ < 0.001) and UK ($\chi^2$ = 32.500, $p$ < 0.001) participants.

= -0.007, $p$ = 0.002); emotional function–fatigue (effect = -0.165, $p$ < 0.001); and emotional function–BMI–fatigue (effect = -0.006, $p$ < 0.001). Female gender therefore predicted worse emotional function, which was associated with higher BMI, and this in turn with greater fatigue. All these variables predicted a lower HRQoL in the participants, which was confirmed

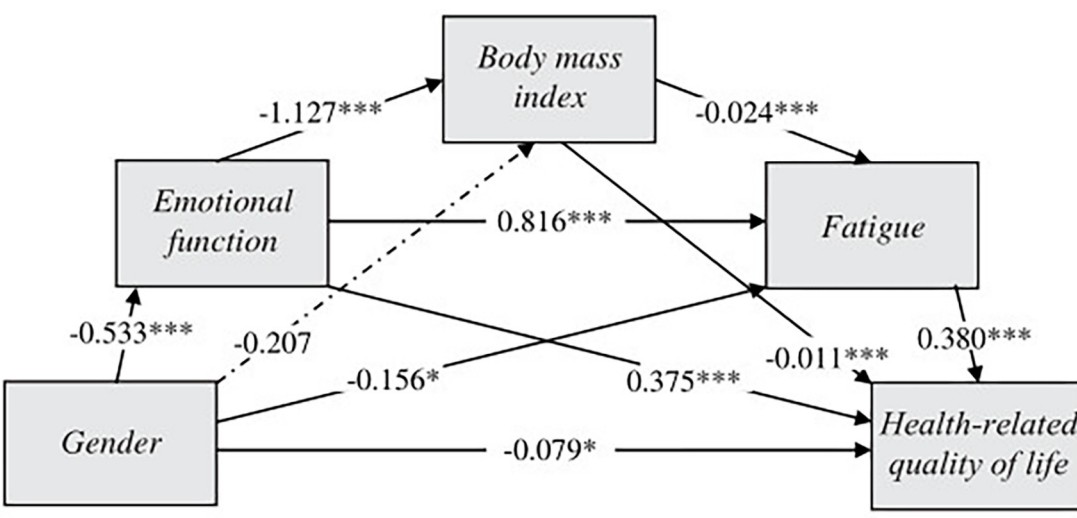

**Fig 3. Emotional function, body mass index and fatigue mediate the relationship between gender and health-related quality of life.** The coefficients represent the indirect and direct effects estimated. *$p$ < 0.05; ***$p$ < 0.001 (mediation and moderated mediation analysis).

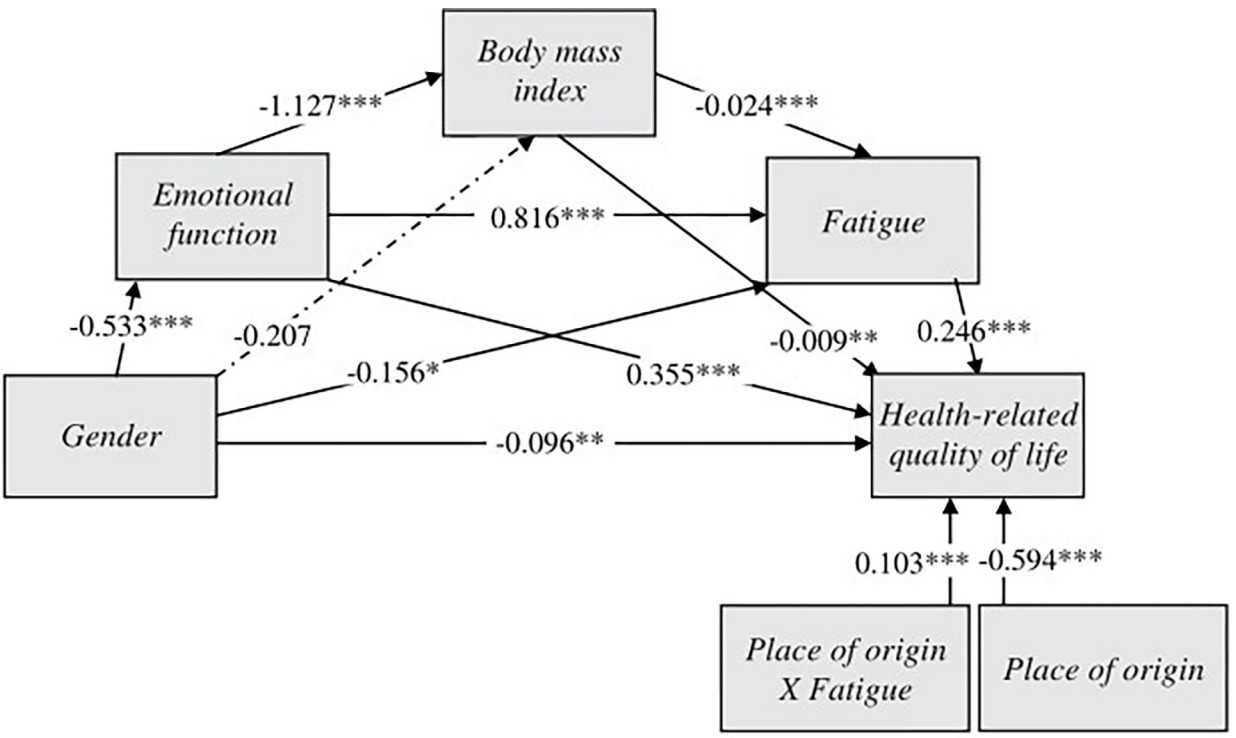

**Fig 4. The moderating effect of place of origin on the relationship between gender and health-related quality of life through emotional function, body mass index and fatigue.** The coefficients represent the moderating, indirect and direct effects estimated. $*p < 0.05$; $**p < 0.01$; $***p < 0.001$ (mediation and moderated mediation analysis).

as the bootstrapped 95% CI did not include 0. Mediation was partial, as the direct effect of gender on HRQoL was significant after mediation analysis (effect = -0.079, $p$ = 0.020).

**2. Moderated mediation model.** Moderated mediation analyses determined whether place of origin moderated the effects of gender on HRQoL through emotional function, BMI and fatigue. The results revealed that place of origin ($\beta$ = 0.103, $p < 0.001$) moderated the relationship between fatigue and HRQoL (Fig 4). The negative effects of fatigue on HRQoL were greater in the UK participants compared to Spanish participants (Spain, effect = 0.349, $p < 0.001$; UK, effect = 0.452, $p < 0.001$) (S2 Table). S3 Table shows the conditional indirect effects of gender on HRQoL through emotional function, BMI and fatigue for the two cohorts. The results showed stronger conditional indirect effects for UK than Spanish participants, with the following significant relationships: emotional function–fatigue (Spain, effect = -0.151, 95% CI = -0.212 to -0.096; UK, effect = -0.196, 95% CI = -0.276 to -0.124); and emotional function–BMI–fatigue (Spain, effect = -0.005, 95% CI = -0.009 to -0.002; UK, effect = -0.007, 95% CI = -0.012 to -0.002). In the pairwise comparisons of conditional indirect effects, the bootstrapped 95% CI did not include 0, confirming mediation moderated by place of origin.

## Discussion

This study analysed the differences in HRQoL for people with NAFLD from two distinct geographical cohorts. The analysis considered the impacts of both cohort and severity of liver damage. Histological and biopsychosocial predictors of HRQoL were also analysed in both cohorts separately. Our analysis also explored whether emotional function, BMI and fatigue

mediated the relationship between gender and HRQoL and whether place of origin moderated that relationship.

There were no important sociodemographic differences between the cohorts, except in degree of liver fibrosis and BMI, which was higher in UK participants as compared to Spanish participants. These differences were expected, considering that the UK leads current estimates of obesity in Europe [30]. Participants with severe fibrosis were older than those with none/mild fibrosis. This result has been reported elsewhere, and is intuitive given it may take time for severe fibrosis to develop [11].

Comparing the two cohorts showed that regardless of their liver severity, the UK participants had lower physical and mental HRQoL, especially with respect to systemic symptoms and worry. This coincides with Huber et al. [12] in emphasizing more deterioration in HRQoL in UK participants, who referred to more physical symptoms, such as body pain or muscular cramps. UK participants reported more nervousness and worry about the evolution of their disease than Spanish participants. It is unclear why this might be the case, although Lazarus et al. [31], concluded that the UK is the European country with the highest level of awareness of NAFLD from a public health policy perspective, whereas Spain had fewer civil society or government strategies for approaching NAFLD. This suggests that our findings may in part be dictated by the relative provision of information and public health messaging between the two countries.

Concerning liver impairment levels, there was no evidence of major differences in HRQoL by absence or presence of NASH regardless whether participants were in the UK or Spanish cohorts. This is similar to the findings of David et al. [10] and Funuyet-Salas et al. [11], but contrary to Huber et al. [12], who suggested that NASH negatively affected HRQoL. However, there were differences in HRQoL in the various levels of fibrosis, where the most important were in the comparison of cirrhotic participants with the none/mild fibrosis group: people with cirrhosis reported more fatigue, systemic symptoms and worry, and a lower HRQoL compared with those with no or mild fibrosis. The decline in HRQoL as symptoms of cirrhosis occur is consistent with previous studies [10,13] including the recent systematic review by McSweeney et al. [8] on HRQoL and PRO measures in NASH-related cirrhosis.

Furthermore, an interaction was found between place of origin and fibrosis for fatigue and HRQoL. Further analysis revealed that UK participants with none/mild fibrosis were more fatigued and had lower HRQoL than Spanish participants. Of the Spanish participants, those who had severe fibrosis showed more fatigue and lower HRQoL than the rest. Spanish participants with moderate fibrosis were also more fatigued than those with none/mild fibrosis. Our study builds on the body of evidence that fibrosis is a predictor of HRQoL of NAFLD participants [10,11]. Having found evidence of differences between levels of none/mild and moderate fibrosis, our analyses demonstrate that the predictive capacity of fibrosis is not due exclusively to damage associated with cirrhosis.

Similar to the reported relationship between mortality and NAFLD [3], the progression and accumulation of fibrosis is a key determinant of the decline in HRQoL observed in these participants. Our work focused on fatigue, which is a recognized persistent dysfunctional problem of NAFLD participants [32]. Fatigue has been associated with neuroinflammation and with altered neurophysiological mechanisms [33,34]. Moreover, the level of cytokeratin 18 (CK18) has been found to correlate positively with patient fatigue [35]. At the same time, higher CK18 serum levels have been found in NAFLD participants than in other chronic liver patients, with a positive association between the levels of this protein and the stage of liver fibrosis [35,36]. Therefore, the action of this type of biomarker could help understand the relationship between HRQoL and fibrosis in NAFLD.

Based on our results, it can also be concluded that fibrosis functioned as a predictor of HRQoL exclusively in the Spanish sample, in which the decline in HRQoL increased as participants advanced toward a cirrhotic state. However, for UK participants, the impact on HRQoL remained without variation over the liver severity levels, with regard to both NASH and liver fibrosis. These results could be interpreted from the perspective of the awareness of the condition mentioned previously. Greater awareness of NAFLD in the UK, with better performance in campaigns for undertaking the disease [31], could be contributing to UK participants identifying their characteristic symptoms better and worrying more about the effect of NAFLD on their health from the first stages of the disease. According to Lazarus et al. [31], the UK is the only European country with multidisciplinary teams and coordination of health professionals in NAFLD management. This could be facilitating better physical and psychological adjustment to the progress of the disease by UK participants than their Spanish counterparts. This would also help explain the differences in how the MELD score predicted HRQoL in Spanish and UK participants, evidence of which has been inconsistent to date [37,38]. A higher MELD score predicted lower HRQoL in Spanish participants for whom severity of liver damage, and specifically, fibrosis, predicts their HRQoL. However, the MELD score was not independently associated with HRQoL in UK participants.

Our results also revealed that the BMI and gender predict HRQoL in both Spanish and UK participants. In line with previous studies [12,16–18], and contradicting the conclusions of Chawla et al. [19] and Sayiner et al. [13], a higher BMI was associated with worse participant HRQoL. Furthermore, female gender was associated with worse HRQoL, a finding reported by others [5,10–12]. As suggested by Huber et al. [12], the CLDQ could show more sensitivity in detecting the negative impact of the disease on women's HRQoL than men's HRQoL.

The inconsistency in the literature on the importance of sociodemographic factors on the HRQoL of NAFLD patients [5,10,11,17,19,20] led us to analyse whether age, education and employment status predicted HRQoL of Spanish and UK participants. Age was positively associated with HRQoL in UK participants, as found in a previous study with NAFLD patients [18]. Keeping in mind that in our study older age was related to higher level of fibrosis, this result would also back the fact that UK participants had better emotional adjustment to the evolution of the disease. On the contrary, education did not predict HRQoL in either Spanish or UK participants, contradicting the results of David et al. [10] and Ozawa et al. [17]. Employment status, on the other hand, was associated with HRQoL in UK participants, where actively employed participants reported better HRQoL than those who were not actively employed, which had already been identified previously in a study on chronic liver pathology [20]. However, employment status did not predict HRQoL in Spanish participants. This could be partly due to the characteristics of the welfare state model in Spain. This model gives an eminent role to the family and formal and informal support networks in the social protection system, which would act as a protective factor for health perception in a non-active or unemployed employment status [39].

Finally, the results of the moderated mediation analysis showed that emotional function, BMI and fatigue partially mediated the relationship between gender and HRQoL. First, female gender predicted worse emotional function, showing female gender to be a major factor contributing to decline in NAFLD patient mental functioning, as previously found by Afendy et al. [5]. Reduced emotional function was associated with higher participant BMI. Worse mental HRQoL has been related to less physical activity and poorer quality diet in terms of less adherence to healthy dietary guidelines in patients with a diversity of chronic pathologies [40–42]. This, in turn, predicts more obesity [43]. Excess fat tends to accumulate mainly in peripheral regions such as the hips or thighs, or in the abdominal cavity, known as central obesity [44]. Patients with central obesity are commonly resistant to insulin, a metabolic condition closely associated with NAFLD and reduced HRQoL, functional capacity and energy [45].

Therefore, higher BMI predicted greater fatigue in our study, which in turn was associated with lower HRQoL. The close relationship between fatigue and HRQoL in NAFLD patients, already identified by Cook et al. [27], was thus confirmed. Place of origin, in turn, moderated this relationship, as the indirect effects of gender on HRQoL through emotional function, BMI and fatigue were higher in UK participants. Therefore, this study found a biopsychosocial risk profile for HRQoL in NAFLD participants, especially those from the UK cohort, based on female gender, poor emotional function, high BMI and greater perception of fatigue.

Intervention to prevent the decline in physical and mental health of patients with an at-risk biopsychosocial profile is especially necessary, considering the decline in HRQoL. NAFLD should therefore be undertaken from a multidisciplinary patient-centered approach [46]. This may prevent some of the greater use of healthcare system resources, lower job productivity and higher mortality these people experience [10]. NAFLD and its impacts should be considered in national and international healthcare policies and be included along with guidelines on clinical management of diabetes, obesity and cardiovascular disease [31].

Our study had some limitations. For example, its cross-sectional design did not enable us to establish causal relationships nor clarify the long-term evolution of the impact of NAFLD on HRQoL. Study participants were diagnosed by liver biopsy, which is the gold standard for the diagnosis and histological assessment of NAFLD [47]. Liver biopsy is part of the standard of care for the diagnosis of NAFLD in both Spanish and UK patient cohorts, which allowed comparison of the data from Spanish and UK participants in this cross-cultural study. Because of its invasive nature, liver biopsy cannot be implemented at early stage and is generally reserved for patients at high risk of advanced liver disease [48]. Study participants may therefore have more impaired HRQoL compared to other studies using non-invasive tests for NAFLD diagnosis. Moreover, other potential effect modifiers such as lifestyle or type 2 diabetes were not considered in the analysis as our comparison can only explore the impact of effect modifiers that are common across both data sets. Nevertheless, the effect of BMI was considered in the analysis, which is relevant as obesity is the main and most common risk factor associated with NAFLD [2]. Future cross-cultural research could analyse the effect of other metabolic comorbidities such as type 2 diabetes or hypertension on HRQoL and could form a focus for future research. In addition, our logistic regression analysis of both data sets included a set of common clinical and sociodemographic effect modifiers (see Statistical Analysis section in the Methods). This allowed us to consider the impact of these potential confounders on HRQoL and draw indirect comparisons between the two cohorts. An alternative approach would have been to draw formal comparisons between groups by constructing a matched cohort. This would have more formally controlled for differences between the two groups in terms of degree of BMI and liver fibrosis. This arguably would have provided a fairer comparison of differences in HRQoL between the two cohorts. It would however have made the analysed 'matched' cohort no longer representative of the population of patients in the two countries i.e. we would have traded external validity for internal validity. A formal matching procedure would also have prevented the indirect exploration of the differential impact of mediating and predictor factors such as liver fibrosis, BMI, age or gender between the two countries (as they would be equalised in a matched cohort). Furthermore, given the difference in the size of the two cohorts, with the UK cohort being approximately one third the size of the Spanish cohort, a matching approach may have reduced our available sample size and hence would have increased the imprecision in our results and so limit our ability to detect the effects of NASH and liver fibrosis on HRQoL. Further studies with larger samples could clarify the clinical and statistical significance of these HRQoL predictors. However, the large size of the study sample, which was comprised of biopsy-proven patients from real clinical practice in Spanish and UK hospitals, constitutes the main strength of this research.

The results of this study showed that HRQoL was mainly lower in UK than Spanish participants, especially in terms of more physical symptoms and worry about the liver disease. Higher fibrosis stage predicted lower HRQoL, mainly in the Spanish cohort. Gender and BMI were found to be independently associated with HRQoL in both Spanish and UK participants. Female gender was associated with worse emotional function, higher BMI and more fatigue, which predicted lower participants' HRQoL. Specifically, the negative impact on NAFLD patients' HRQoL was greater in UK than in Spanish participants. Our results confirm and extend knowledge of the impact of NAFLD from the individual's perspective. This cross-cultural study will enable healthcare professionals to better understand the biopsychosocial factors that predict and contribute to the impact of NAFLD on patient HRQoL, as well as identify important differences in HRQoL of Spanish and UK patients with this liver disease.

## Supporting information

**S1 Table. Indirect effects of emotional function, body mass index and fatigue mediating in the association between gender and health-related quality of life.**
(DOCX)

**S2 Table. Effects of moderation by place of origin (Spain or UK) on the relationship between fatigue and health-related quality of life.**
(DOCX)

**S3 Table. Conditional indirect effect of gender (male and female) on health-related quality of life through emotional function, body mass index and fatigue.**
(DOCX)

## Acknowledgments

The authors want to thank the patients for their participation.

## Author Contributions

**Conceptualization:** Jesús Funuyet-Salas, Agustín Martín-Rodríguez, María Ángeles Pérez-San-Gregorio, Luke Vale, Tomos Robinson, Quentin M. Anstee, Manuel Romero-Gómez.

**Data curation:** Jesús Funuyet-Salas.

**Formal analysis:** Jesús Funuyet-Salas, Agustín Martín-Rodríguez, María Ángeles Pérez-San-Gregorio, Luke Vale, Tomos Robinson, Quentin M. Anstee, Manuel Romero-Gómez.

**Funding acquisition:** Agustín Martín-Rodríguez, María Ángeles Pérez-San-Gregorio, Quentin M. Anstee, Manuel Romero-Gómez.

**Investigation:** Jesús Funuyet-Salas, Agustín Martín-Rodríguez, María Ángeles Pérez-San-Gregorio, Luke Vale, Tomos Robinson, Quentin M. Anstee, Manuel Romero-Gómez.

**Methodology:** Jesús Funuyet-Salas, Agustín Martín-Rodríguez, María Ángeles Pérez-San-Gregorio, Luke Vale, Tomos Robinson, Quentin M. Anstee, Manuel Romero-Gómez.

**Project administration:** Agustín Martín-Rodríguez, María Ángeles Pérez-San-Gregorio, Luke Vale, Quentin M. Anstee, Manuel Romero-Gómez.

**Resources:** Agustín Martín-Rodríguez, María Ángeles Pérez-San-Gregorio, Luke Vale, Tomos Robinson, Quentin M. Anstee, Manuel Romero-Gómez.

**Software:** Jesús Funuyet-Salas, Agustín Martín-Rodríguez, María Ángeles Pérez-San-Gregorio, Luke Vale, Tomos Robinson, Quentin M. Anstee, Manuel Romero-Gómez.

**Supervision:** Agustín Martín-Rodríguez, María Ángeles Pérez-San-Gregorio, Luke Vale, Tomos Robinson, Quentin M. Anstee, Manuel Romero-Gómez.

**Validation:** Jesús Funuyet-Salas, Agustín Martín-Rodríguez, María Ángeles Pérez-San-Gregorio, Luke Vale, Tomos Robinson, Quentin M. Anstee, Manuel Romero-Gómez.

**Visualization:** Jesús Funuyet-Salas, Agustín Martín-Rodríguez, María Ángeles Pérez-San-Gregorio, Luke Vale, Tomos Robinson, Quentin M. Anstee, Manuel Romero-Gómez.

**Writing – original draft:** Jesús Funuyet-Salas.

**Writing – review & editing:** Jesús Funuyet-Salas, Agustín Martín-Rodríguez, María Ángeles Pérez-San-Gregorio, Luke Vale, Tomos Robinson, Quentin M. Anstee, Manuel Romero-Gómez.

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
