## [Decision Letter · Decision Letter 0]

6 Jan 2023

PONE-D-22-34276Health-related quality of life in non-alcoholic fatty liver disease: A cross-cultural study between Spain and the United KingdomPLOS ONE

Dear Dr. Funuyet-Salas,

Thank you for submitting your manuscript to PLOS ONE. After careful consideration, we feel that it has merit but does not fully meet PLOS ONE’s publication criteria as it currently stands. Therefore, we invite you to submit a revised version of the manuscript that addresses the points raised during the review process.

Dear Dr. Jesús Funuyet-Salas, this manuscript was reviewed by 2 independent reviewers and the editor. The major concerns were the lack of matching between the two patients’ groups (Spanish and UK) regarding BMI, education level and employment activity.  In addition, the two patient groups were not compared regarding the severity of liver fibrosis and the presence of associated comorbidities (like diabetes mellitus, hypertension, etc..). Indeed, these concerns need major revision, please review the reviewers' comments.  

We look forward to receiving your revised manuscript.

Kind regards,

Ashraf Elbahrawy

Academic Editor

PLOS ONE

Additional Editor Comments:

Dear Dr. Jesús Funuyet-Salas, this manuscript was reviewed by 2 independent reviewers and the editor. The major concerns were the lack of matching between the two patients’ groups (Spanish and UK) regarding BMI, education level and employment activity. In addition, the two patient groups were not compared regarding the severity of liver fibrosis and the presence of associated comorbidities (like diabetes mellitus, hypertension, etc..). Indeed, these concerns need major revision, please review the reviewers' comments.

Reviewers' comments:

Reviewer's Responses to Questions

**Comments to the Author**

1. Is the manuscript technically sound, and do the data support the conclusions?

Reviewer #1: Yes

Reviewer #2: Yes

2. Has the statistical analysis been performed appropriately and rigorously? 

Reviewer #1: Yes

Reviewer #2: Yes

3. Have the authors made all data underlying the findings in their manuscript fully available?

Reviewer #1: Yes

Reviewer #2: No

4. Is the manuscript presented in an intelligible fashion and written in standard English?

Reviewer #1: Yes

Reviewer #2: No

5. Review Comments to the Author

Reviewer #1: Title is very relevant for holistic management of NAFLD as HRQoL is underestimated issue.

Introduction covers background knowledge, knowledge gap and specific objectives of the study clearly.

In Methodology patient reported outcome CLDQ followed to search HRQoL but need to exclude other chronic functional illness associated with NAFLD eg diabetes, hypertension and these may be confounder. These confounding effect was not removed

Definition of NAFLD done only Biopsy, possibly these patients are more advanced NAFLD so quality of life more disrupted

Result and abstract of manuscript were concordant.

Reviewer #2: Thanks for inviting me to review this manuscript.

Funuyet-Salas et al., described Health-related quality of life in non-alcoholic fatty liver disease patients from Spain and the United Kingdom.

I have the following comments:

1- Although interesting, however it is not novel. The same research question was addressed in a similar study from different European countries including Spain and UK (https://pubmed.ncbi.nlm.nih.gov/30580090/ ). Another recent review confirmed the same (https://pubmed.ncbi.nlm.nih.gov/32435754/ ).

2- The manuscript needs language editing (e.g., “Methods: HRQoL (CLDQ) was measure in both Southern” should be written as “measured”).

3- In the sentence “In general, the negative impact of MODERATORS on Harmol was reported to a greater degree in UK than in Spanish participants” what does moderator means?

4- Page 10 line 99 the expression “and liver severity” is not appropriate “liver diseases severity” is more suitable.

5- It is preferred to add S1 and S2 Tables to the main document.

6- Please add ethics committee approval number

7- Page 13 line 189: the sentence “There were no important between-group differences” needs rephrasing.

6. PLOS authors have the option to publish the peer review history of their article (what does this mean?). If published, this will include your full peer review and any attached files.

Reviewer #1: **Yes: **Shahinul Alam

Reviewer #2: **Yes: **Prof. Mohamed Alboraie

---

## [Author Response · Author response to Decision Letter 0]

17 Feb 2023

Response to Editor Comments

Point 1:

Dear Dr. Jesús Funuyet-Salas, this manuscript was reviewed by 2 independent reviewers and the editor. The major concerns were the lack of matching between the two patients’ groups (Spanish and UK) regarding BMI, education level and employment activity. In addition, the two patient groups were not compared regarding the severity of liver fibrosis and the presence of associated comorbidities (like diabetes mellitus, hypertension, etc..). Indeed, these concerns need major revision, please review the reviewers' comments.

Response 1: 

Dear editor, thank you for your review. The lack of matching between the two patient cohorts regarding BMI and liver fibrosis was discussed as a study limitation. Differences in education and employment were statistically significant but not important (small effect sizes). Effect sizes were defined as: null (d < 0.2; w < 0.1), small (d > 0.2; w > 0.1), medium (d > 0.5; w > 0.3) or large (d > 0.8; w > 0.5). Only statistically significant differences with medium or large effect sizes were considered important in this manuscript. This is discussed in Page 8 lines 158-160.

In addition, the two patient cohorts were compared with respect to BMI as obesity is the metabolic condition most closely linked to NAFLD. The fact that we did not consider the effects of other metabolic comorbidities such as diabetes or hypertension has been discussed as a study limitation in Page 23 line 420. We hope the editor finds this explanation sufficient.

 

Response to Reviewer 1 Comments

Point 1:

Title is very relevant for holistic management of NAFLD as HRQoL is underestimated issue.

Introduction covers background knowledge, knowledge gap and specific objectives of the study clearly.

Response 1: 

Thank you for your review and your encouraging comments on our manuscript. 

Point 2:

In Methodology patient reported outcome CLDQ followed to search HRQoL but need to exclude other chronic functional illness associated with NAFLD eg diabetes, hypertension and these may be confounder. These confounding effect was not removed.

Response 2: 

Thank you for this suggestion. The two patient cohorts were compared with respect to BMI. This is because obesity is the metabolic condition most closely linked to NAFLD. The fact that we did not consider the effects of other metabolic comorbidities such as diabetes or hypertension was discussed as a study limitation in Page 23 line 420.

Point 3:

Definition of NAFLD done only Biopsy, possibly these patients are more advanced NAFLD so quality of life more disrupted.

Response 3: 

Thank you for your comment. Liver biopsy is currently considered as the gold standard for the diagnosis and histological assessment of NAFLD. All the participants were biopsy-proven NAFLD patients, which provides added value to the validity of the study results. In terms of liver severity, we recruited patients with no or mild fibrosis as well as those with moderate or severe fibrosis. However, this has been discussed as a study limitation in Page 23 line 419.

Point 4:

Result and abstract of manuscript were concordant.

Response 4: 

Thank you for your positive comment on our manuscript. 

Response to Reviewer 2 Comments

Point 1:

Funuyet-Salas et al., described Health-related quality of life in non-alcoholic fatty liver disease patients from Spain and the United Kingdom. I have the following comments: 1- Although interesting, however it is not novel. The same research question was addressed in a similar study from different European countries including Spain and UK (https://pubmed.ncbi.nlm.nih.gov/30580090/). Another recent review confirmed the same (https://pubmed.ncbi.nlm.nih.gov/32435754/).

Response 1: 

Thank you for your review and your precise helpful suggestions that contributed to the improvement of the manuscript.

As suggested in the introduction of the manuscript, the study by Huber et al., (2019) is the only one that has so far compared the quality of life of NAFLD patients in different European countries. Indeed, whilst the study included Spain, it only reported on 17 participants from Spain. Our manuscript is substantially larger and included 513 Spanish participants. We therefore believe that our study provides more accurate results on the comparison between Spain and other countries. In addition, cross-cultural comparisons were only addressed minor component of Huber et al., (2019). In order to understand this issue better, we sought to go deeper into the differences in HRQoL between Spanish and UK participants as is described in Page 4 line 98. 

For McSweeney et al., (2020), which was led by members of our team was a review of extant literature, it focused mainly on the impact of NASH-associated cirrhosis from a humanistic perspective. The literature it considered whilst useful was very different in term of design to the current study and was designed to support the development of a patient reported outcome measure suitable for use in trials of treatments for NAFLD. The study was not in any way quantitative and sought to identify issues and themes relevant to the assessment of HRQoL. Given our focus on HRQoL in NAFLD patients from a cross-cultural and quantitative approach we strongly argue that the current study and McSweeney et al., (2020) do not overlap at all but are complementary in terms of adding to the evidence base for NAFLD.

Point 2:

2- The manuscript needs language editing (e.g., “Methods: HRQoL (CLDQ) was measure in both Southern” should be written as “measured”).

Response 2: 

Thank you for this correction. Change made.

Point 3:

3- In the sentence “In general, the negative impact of MODERATORS on HRQoL was reported to a greater degree in UK than in Spanish participants” what does moderator means?

Response 3: 

"Moderator" means the mediating and moderating variables in the moderated mediation model (third study objective). Specifically, the negative effects of gender on quality of life through emotional function, BMI and fatigue were higher in UK than in Spanish participants. This has been edited to put the text in plain language (Page 2 line 42). 

Point 4:

4- Page 10 line 99 the expression “and liver severity” is not appropriate “liver diseases severity” is more suitable.

Response 4: 

Thank you for this correction. Change made.

Point 5:

5- It is preferred to add S1 and S2 Tables to the main document.

Response 5: 

Thank you for this suggestion. Change made.

Point 6:

6- Please add ethics committee approval number.

Response 6: 

Thank you. We have now added this information.

Point 7:

7- Page 13 line 189: the sentence “There were no important between-group differences” needs rephrasing.

Response 7: 

Thank you for this suggestion. Change made.

---

## [Decision Letter · Decision Letter 1]

22 Mar 2023

PONE-D-22-34276R1

Health-related quality of life in non-alcoholic fatty liver disease: A cross-cultural study between Spain and the United Kingdom

PLOS ONE

Dear Dr. Funuyet-Salas,

Thank you for submitting your manuscript to PLOS ONE. After careful consideration, we have decided that your manuscript does not meet our criteria for publication and must therefore be rejected.

I am sorry that we cannot be more positive on this occasion, but hope that you appreciate the reasons for this decision.

Kind regards,

Ashraf Elbahrawy

Academic Editor

PLOS ONE

Additional Editor Comments:

Dear Dr. Jesús Funuyet-Salas

There are still some major issues related to the lack of matching between groups and the absence of comparison regarding some potential confounding factors that might affect the final results. Please revise the reviewers comments

Reviewers' comments:

Reviewer's Responses to Questions

**Comments to the Author**

1. If the authors have adequately addressed your comments raised in a previous round of review and you feel that this manuscript is now acceptable for publication, you may indicate that here to bypass the “Comments to the Author” section, enter your conflict of interest statement in the “Confidential to Editor” section, and submit your "Accept" recommendation.

Reviewer #3: All comments have been addressed

Reviewer #4: (No Response)

2. Is the manuscript technically sound, and do the data support the conclusions?

Reviewer #3: Yes

Reviewer #4: Partly

3. Has the statistical analysis been performed appropriately and rigorously? 

Reviewer #3: Yes

Reviewer #4: N/A

4. Have the authors made all data underlying the findings in their manuscript fully available?

Reviewer #3: Yes

Reviewer #4: Yes

5. Is the manuscript presented in an intelligible fashion and written in standard English?

Reviewer #3: Yes

Reviewer #4: Yes

6. Review Comments to the Author

Reviewer #3: Patient-reported outcome is an important but often neglected topic. The authors have addressed previous Reviewers' comments adequately. I do not have anything else to add.

Reviewer #4: There are still some major issues related to the lack of matching between the Spanish and the UK groups, the absence of comparison regarding some potential confounding factors such as diabetes mellitus, and other stated limitations that might affect the results and the resulting conclusions.

7. PLOS authors have the option to publish the peer review history of their article (what does this mean?). If published, this will include your full peer review and any attached files.

Reviewer #3: No

Reviewer #4: No

- - - - -

---

## [Author Response · Author response to Decision Letter 1]

16 May 2023

Response to Academic Editor Comments:

Dear Academic Editor, thank you for your review and the decision to reconsider our manuscript. All changes to the manuscript have been highlighted in yellow. 

The lack of matching between the two patient cohorts regarding BMI and liver fibrosis has now been extensively discussed (page 24, lines 429-444). We understand that this could affect the validity of our results. We argue that although a formal matching procedure (using a technique such as propensity score matching) has not been conducted, we did use a consistent regression analytical framework across both data sets. This allowed us to highlight the impact of a common set of effect modifiers which included BMI and fibrosis. If the aim of our paper was to compare differences in quality of life between countries, then a formal matching procedure would have been appropriate. However, the aim of our paper was to specifically look at the mediating and predictor factors. A formal matching procedure would have prevented the indirect exploration of the differential impact of mediating and predictor factors such as liver fibrosis, BMI, age or gender between the two countries (as they would be equalised in a matched cohort). The analysis as it stands allows us to consider if these confounders could have a differential impact between countries.

As noted above, the two patient cohorts were compared with respect to BMI, with obesity being the most common metabolic condition in NAFLD patients. As the reviewers have noted this approach will not have considered other potential confounding factors such as diabetes, as it can only explore the impact of effect modifiers that are consistent across both data sets. Nevertheless, several potential confounders were consistently explored in the two data sets (page 8, lines 170-175). The fact that we did not consider the effects of other metabolic comorbidities such as diabetes or hypertension has been discussed as a study limitation (page 24, lines 424-429) and highlighted as an area for further research. 

We hope the editor finds this explanation sufficient.

Response to Reviewer 3 Comments:

Thank you for your review and your positive comment. 

Response to Reviewer 4 Comments:

Thank you for your review and your comments.

Reviewer 4 raised issues relating to the control of BMI and fibrosis. We cross refer to our response to the editor above on these issues. We trust that reviewer 4 finds this response satisfactory.

---

## [Decision Letter · Decision Letter 2]

11 Oct 2023

PONE-D-22-34276R2

Health-related quality of life in non-alcoholic fatty liver disease: A cross-cultural study between Spain and the United Kingdom

PLOS ONE

Dear Dr. Funuyet-Salas,

Thank you for submitting your manuscript to PLOS ONE. After careful consideration, we feel that it has merit but does not fully meet PLOS ONE’s publication criteria as it currently stands. Therefore, we invite you to submit a revised version of the manuscript that addresses the points raised during the review process.

ACADEMIC EDITOR: Please insert comments here and delete this placeholder text when finished. Be sure to:

Indicate which changes you require for acceptance versus which changes you recommendAddress any conflicts between the reviews so that it's clear which advice the authors should followProvide specific feedback from your evaluation of the manuscript

We look forward to receiving your revised manuscript.

Kind regards,

Mohamed El-Kassas

Academic Editor

PLOS ONE

Journal Requirements:

1. We notice that your manuscript file was uploaded on February 17, 
2023. Please can you upload the latest version of your revised manuscript as the main article file, ensuring that does not contain any tracked changes or highlighting. This will be used in the production process if your manuscript is accepted. Please follow this link for more information: http://blogs.PLOS.org/everyone/2011/05/10/how-to-submit-your-revised-manuscript/

Additional Editor Comments (if provided):

Reviewers' comments:

Reviewer's Responses to Questions

**Comments to the Author**

1. If the authors have adequately addressed your comments raised in a previous round of review and you feel that this manuscript is now acceptable for publication, you may indicate that here to bypass the “Comments to the Author” section, enter your conflict of interest statement in the “Confidential to Editor” section, and submit your "Accept" recommendation.

Reviewer #3: All comments have been addressed

Reviewer #5: All comments have been addressed

2. Is the manuscript technically sound, and do the data support the conclusions?

Reviewer #3: Yes

Reviewer #5: Yes

3. Has the statistical analysis been performed appropriately and rigorously? 

Reviewer #3: Yes

Reviewer #5: Yes

4. Have the authors made all data underlying the findings in their manuscript fully available?

Reviewer #3: Yes

Reviewer #5: Yes

5. Is the manuscript presented in an intelligible fashion and written in standard English?

Reviewer #3: Yes

Reviewer #5: Yes

6. Review Comments to the Author

Reviewer #3: 6. Review Comments to the Author

Please use the space provided to explain your answers to the questions above. You may also include additional comments for the author, including concerns about dual publication, research ethics, or publication ethics.

Response: I do not have further suggestions.

Reviewer #5: 1-Methodology: You made the NAFLD diagnosis by taking liver biopsy. Why did you do liver biopsy? Is it for your study or indicated otherwise? it should be addressed in the manuscript.

2- their was a statistical significant difference between the 2 groups regarding their education level and occupation. As you know, the socioeconomic level of the patients could greatly affect their HRQoL. How could you prove that the difference you found in the HRQoL is not due to the difference in socioeconomic level?

3-You chose Spain and UK as a distinct geographical areas. Do you think they are distinct? why didn't you choose 2 so far areas especially from 2 different continents?

4. Please consider grammatical reversion of some errors.

7. PLOS authors have the option to publish the peer review history of their article (what does this mean?). If published, this will include your full peer review and any attached files.

Reviewer #3: **Yes: **Vincent Wong

Reviewer #5: **Yes: **Mohamed Elbadry

---

## [Author Response · Author response to Decision Letter 2]

19 Oct 2023

Response to Reviewer 3 Comments

Point 1:

I do not have further suggestions.

Response 1:

Thank you for your review that contributed to the improvement of the manuscript. 

Response to Reviewer 5 Comments

Point 1:

Methodology: You made the NAFLD diagnosis by taking liver biopsy. Why did you do liver biopsy? Is it for your study or indicated otherwise? it should be addressed in the manuscript.

Response 1:

Thank you for your review and your helpful suggestions that contributed to the improvement of the manuscript. The 513 Spanish participants were part of Hepamet, the Spanish registry, which diagnoses NAFLD by liver biopsy. The 224 UK participants were part of the European NAFLD Registry, which also uses liver biopsy to diagnose NAFLD. 

Liver biopsy is part of the standard of care for the diagnosis of NAFLD in both Spanish and UK patient cohorts, which allowed comparison of the data from Spanish and UK participants in this cross-cultural study. This has been addressed in the manuscript (page 24, lines 423-425).

Point 2:

There was a statistical significant difference between the 2 groups regarding their education level and occupation. As you know, the socioeconomic level of the patients could greatly affect their HRQoL. How could you prove that the difference you found in the HRQoL is not due to the difference in socioeconomic level?

Response 2:

Thank you for this comment. There was indeed a statistically significant difference between the two groups in terms of education and employment. However, the effect sizes were not important (w = 0.194 and -0.102, respectively), as you can see in Table 1. Effect sizes are defined as: null (d < 0.2; w < 0.1), small (d > 0.2; w > 0.1), medium (d > 0.5; w > 0.3) or large (d > 0.8; w > 0.5). Only statistically significant differences with medium or large effect sizes were considered important in this manuscript (page 8, lines 158-160).

Point 3:

You chose Spain and UK as distinct geographical areas. Do you think they are distinct? why didn't you choose 2 so far areas especially from 2 different continents?

Response 3:

The aim of this study was to compare and explore quality of life in both Southern European and Northern European cohorts of patients with NAFLD. For this reason, Spain and the United Kingdom were selected for this cross-cultural study. We believe that both countries are different not only in relation to lifestyle, socio-cultural issues or estimates of obesity, but also in terms of awareness and strategies for approaching NAFLD from a public health policy perspective, as suggested in the manuscript (see Lazarus et al., 2021). 

Point 4:

Please consider grammatical reversion of some errors.

Response 4:

Thank you for this comment. The manuscript has been grammatically revised again. All changes have been highlighted in yellow.

---

## [Decision Letter · Decision Letter 3]

27 Feb 2024

Health-related quality of life in non-alcoholic fatty liver disease: A cross-cultural study between Spain and the United Kingdom

PONE-D-22-34276R3

Dear Dr. Funuyet-Salas,

We’re pleased to inform you that your manuscript has been judged scientifically suitable for publication and will be formally accepted for publication once it meets all outstanding technical requirements.

Kind regards,

Matias A Avila, Ph.D.

Academic Editor

PLOS ONE

Additional Editor Comments (optional):

Reviewers' comments:

Reviewer's Responses to Questions

**Comments to the Author**

1. If the authors have adequately addressed your comments raised in a previous round of review and you feel that this manuscript is now acceptable for publication, you may indicate that here to bypass the “Comments to the Author” section, enter your conflict of interest statement in the “Confidential to Editor” section, and submit your "Accept" recommendation.

Reviewer #3: All comments have been addressed

2. Is the manuscript technically sound, and do the data support the conclusions?

Reviewer #3: Yes

3. Has the statistical analysis been performed appropriately and rigorously? 

Reviewer #3: Yes

4. Have the authors made all data underlying the findings in their manuscript fully available?

Reviewer #3: Yes

5. Is the manuscript presented in an intelligible fashion and written in standard English?

Reviewer #3: Yes

6. Review Comments to the Author

Reviewer #3: Thanks a lot for your thoughtful revision. I do not have anything else to add. Congratulations on the publication!

7. PLOS authors have the option to publish the peer review history of their article (what does this mean?). If published, this will include your full peer review and any attached files.

Reviewer #3: No
